# Differential Predictors of Response to Early Start Denver Model vs. Early Intensive Behavioral Intervention in Young Children with Autism Spectrum Disorder: A Systematic Review and Meta-Analysis

**DOI:** 10.3390/brainsci12111499

**Published:** 2022-11-04

**Authors:** Lisa Asta, Antonio M. Persico

**Affiliations:** Child & Adolescent Neuropsychiatry Program, Modena University Hospital, Department of Biomedical, Metabolic and Neural Sciences, University of Modena and Reggio Emilia, 41125 Modena, Italy

**Keywords:** applied behavioral analysis, autism spectrum disorder, developmental quotient, early intervention, early intensive behavioral intervention, early start denver model, naturalistic behavioral developmental intervention, predictors, treatment outcome

## Abstract

The effectiveness of early intensive interventions for Autism Spectrum Disorder (ASD) is now well-established, but there continues to be great interindividual variability in treatment response. The purpose of this systematic review is to identify putative predictors of response to two different approaches in behavioral treatment: Early Intensive Behavioral Interventions (EIBI) and the Early Start Denver Model (ESDM). Both are based upon the foundations of Applied Behavioral Analysis (ABA), but the former is more structured and therapist-driven, while the latter is more naturalistic and child-driven. Four databases (EmBase, PubMed, Scopus and WebOfScience) were systematically screened, and an additional search was conducted in the reference lists of relevant articles. Studies were selected if participants were children with ASD aged 12–48 months at intake, receiving either EIBI or ESDM treatment. For each putative predictor, *p*-values from different studies were combined using Fisher’s method. Thirteen studies reporting on EIBI and eleven on ESDM met the inclusion criteria. A higher IQ at intake represents the strongest predictor of positive response to EIBI, while a set of social cognitive skills, including intention to communicate, receptive and expressive language, and attention to faces, most consistently predict response to ESDM. Although more research will be necessary to reach definitive conclusions, these findings begin to shed some light on patient characteristics that are predictive of preferential response to EIBI and ESDM, and may provide clinically useful information to begin personalizing treatment.

## 1. Introduction

Autism Spectrum Disorder (ASD) is a heterogeneous disorder characterized by persistent deficits in social communication and interaction, by repetitive behaviors, and restricted interests or activities, and abnormal sensory processing. Moreover, these children often display co-morbid intellectual disability and language impairment [1]. Genetics strongly contribute to ASD, as supported by concordance in monozygotic twins being consistently higher than that observed in dizygotic twins [2]. A specific genetic aetiology is identifiable in up to 40% of individuals, although the majority of cases display complex gene–gene interactions involving multiple common and rare variants [3,4,5]. For many patients, also gene–environment interactions involving a genetic predisposition and prenatal–early-postnatal environmental influences are also plausible [6]. In addition to diagnosis, genetic variants can also contribute to explain interindividual variability in clinical phenotype, developmental trajectories, and responsiveness to behavioral or pharmacological treatment [7,8]. Hence, heterogeneity at the pathogenetic level translates into great clinical and treatment-related interindividual differences.

To date, there is no medical or biological treatment for core ASD symptoms, and interventions for ASD mainly fall within the psychoeducational, psychosocial, or behavioral frameworks [9,10]. Furthermore, there is no standard treatment for ASD. Community mental health programs, based on local guidelines, typically include a combination of interventions based on various approaches, such as speech therapy, sensory integration therapy, occupational therapy, neuropsychomotor treatment, and so on. However, these approaches are of limited efficacy and are not always evidence-based, so more structured and comprehensive interventions, such as those derived from Applied Behavioral Analysis (ABA), should be preferred [9].

Over the last 40 years, treatments based on ABA have become increasingly popular, especially after the publication of Lovaas’ promising results in 1987 [11]. In this now famous study, Lovaas showed that half of 19 children undergoing a manualized early intensive behavioral intervention (EIBI) [12], mainly based on discrete trial teaching (DTT), achieved normal cognitive and educational functioning as compared to children receiving less intensive behavioral intervention or other types of treatment. Since then, studies on EIBI have proliferated, confirming the effectiveness of Early Intensive Behavioral Intervention, although no study has replicated results as favorable as those initially reported by Lovaas and colleagues [13,14,15,16,17].

Although effective in teaching new skills, research has shown that highly structured interventions such as DTT may have limitations, such as children having difficulties in generalizing learned skills in different contexts [18]. These limitations have led to the design and implementation of new approaches to behavioral intervention, still based on ABA, but less structured and with more naturalistic features: the Naturalistic Developmental Behavioral Interventions [19]. One type of intervention belonging to this family is the Early Start Denver Model (ESDM) [19], a comprehensive early intervention designed for children aged 12–48 months, whose efficacy and effectiveness have now been proven by several studies, including randomized controlled trials (RCT) and meta-analysis [20,21,22,23,24]. 

Although research on early and comprehensive behavioral interventions, such as EIBI and ESDM, has shown that children can achieve optimal outcomes, studies have found great interindividual variability in rate and extent of clinical improvement, with some children showing larger gains and others only small progress (e.g., [25,26,27,28]). This variability is not surprising, given the great heterogeneity in etiopathogenetic underpinnings, developmental trajectories, symptom patterns and severity present in the “Autisms”, term used by many investigators to comprehensively refer to ASD as a heterogeneous collection of rare disorders sharing the clinical features defined as “diagnostic criteria” in DSM-5 [29]. Within this framework, researchers have attempted to identify factors that may be associated with more favorable spontaneous developmental trajectories in children at risk or initially diagnosed with ASD. The most frequently reported factors are baseline cognitive abilities and severity of autism symptoms. For example, Weismer and Kover [30] found that ASD symptom severity and cognitive abilities at 30 months were significant predictors of language development at 66 months, in 129 children mostly receiving behavioral interventions at the time of the final assessment. Instead, fine motor skills, and not nonverbal cognitive abilities, were positive predictors of expressive language development in two independent samples of 86 and 181 children, assessed at age 3 and again at age 19 or 10.5, respectively [31]. Other factors, such as imitation and joint attention, were reported to be predictive of a favorable developmental trajectory, but not consistently in all studies. 

In addition to interindividual variability, differences in treatment methodology could produce a more or less favorable outcome in different subgroups of autistic children. This notion spurred interest in searching for predictors of a positive response to specific forms of early intensive intervention. Two studies by Schreibman and Colleagues [32,33] appear to be especially interesting, as they suggest that the child characteristics associated with treatment outcome may be related to the style of treatment delivered. In their first study [32], the Authors identified two distinct behavioral profiles at baseline for responders and non-responders to a NDBI intervention, the Pivotal Response Training (PRT). Initially, the Authors predicted that children responding to PRT would show more toy play, less social avoidance and more verbal self-stimulatory behaviors, and this prediction was indeed proven correct [32]. In their subsequent study [33], the Authors selected six children with an incomplete “responder” profile, as three children lacked high toy play and three other children lacked low social avoidance, in the presence of the other two predictors. These six children received PRT first, and then DTT. PRT produced no significant response in children lacking only toy play, while children lacking only social avoidance displayed intermediate improvements between those of “responders” and “non responders” in the original study [32], pointing toward a greater role for toy play in predicting response to PRT. Importantly, the “PRT responder” profile did not predict response to DTT [33]. These two studies, for the first time, pointed to the existence of different sets of predictors of response to different forms of behavioral treatment, lending support to the possible personalization of early intervention in newly diagnosed ASD children.

Several studies have investigated factors that may be specifically associated with positive outcomes to EIBI and ESDM, yielding variable results. Higher intellectual functioning, measured as IQ or DQ at intake, was among the strongest predictors of response to EIBI in many [11,34,35,36,37], but not all, studies [13,16,23]. Starting treatment as early as possible seems to positively affect intervention outcome, and experts recommend that children should be referred to behavioral interventions as soon as ASD is diagnosed [10,38]. However, only some studies found that a younger age at intake predicted better outcome [36,39,40,41]. For example, Lovaas [11] did not find younger age at treatment intake to be associated with the best outcomes. Pretreatment autism severity and language skills have also been reported to predict treatment outcome, again with mixed results (positive for EIBI [36,42,43]; negative for EIBI [39,44,45]; positive for ESDM [27,46]; negative for ESDM [22,23,47]). Finally, other skills were found to be associated with positive outcomes, such as imitation [28,48] and joint attention [49,50], but these were not confirmed in other studies [22,45,51]. 

Though EIBI and ESDM seem to be equally effective in improving children outcome [23], their curriculum and teaching methods are indeed different. Therefore, it is plausible that some children are more likely to respond to one treatment approach than the other, based on their underlying neurobiology and genetics, which may express a set of clinically observable pre-treatment characteristics. Furthermore, the variable age range of children recruited in prior studies may have contributed to their discordant results, because greater deficits in a given function at an older age may reflect a more impaired underlying neurobiology, as compared to younger children, whose developmental trajectory is still at an earlier stage. This study has two main aims: (1) to systematically review all the available literature on predictors of response to two different types of behavioral interventions, notably Early Intensive Behavioral Intervention and Early Start Denver Model, in young children diagnosed with ASD and whose treatment starts by 48 months of age; (2) to combine evidence from different studies to define first- and second-line predictors of outcome for each intervention method based on the available evidence. Focusing on studies recruiting only young children should partly reduce inconsistencies and provide more helpful indications in the clinic. In fact, knowing which factors are most associated with a better response to treatment in this early stage of development, and whether these factors are treatment-specific, could help clinicians to prescribe the most effective intervention for each single child, at the time in life in which neuronal plasticity is at its maximum. 

## 2. Methods

### 2.1. Design and Data Sources

Studies included in this systematic review were identified through a search performed on the following databases: EmBase, PubMed, Scopus and WebOfScience (WOS) [date of search: 9 September 2022]. 

Our search string was as follows: (autism OR autism spectrum disorder OR asd) AND (predictor OR predicting outcome OR outcome) AND (early intervention OR early start denver model OR esdm OR early intensive behavioral intervention OR eibi).

### 2.2. Study Inclusion and Exclusion Criteria

Only quantitative, empirical studies published in peer-reviewed journals were included. Studies were selected if participants were very young children under the age of four years at patient intake (i.e., 12–48 months); meeting DSM-5 criteria for Autism Spectrum Disorder [1], or DSM-III/DSM-IV criteria for Autistic Disorder and/or Pervasive Developmental Disorder—Not Otherwise Specified (PDD-NOS) [52,53], or ICD-10 criteria for Autistic Disorder [54]; receiving either Early Start Denver Model or Early Intensive Behavioral Intervention applied by a certified therapist. Studies were excluded if they focused on neurodevelopmental disorders of known genetic etiology (e.g., Fragile-X Syndrome, Rett Syndrome, Tuberous Sclerosis Complex); did not report pre-treatment child characteristics as predictors of ESDM/EIBI outcome; or applied parent-mediated interventions, whereby one parent was the main therapist. 

### 2.3. Assessment and Measures

Overall, our initial search yielded 1601 articles, including 475 in WOS, 212 in PubMed, 666 in Scopus, and 248 in Embase. Articles were screened for eligibility based on title, abstract and, when appropriate, full text. We focused on children characteristics as potential predictors of treatment outcome, including anagraphical data and developmental measures, such as chronological age, cognitive abilities, language skills, and autism symptoms severity, recorded at the start of treatment. 

### 2.4. Study Selection Process

After removing duplicates from the different databases, 1121 articles were identified; 1107 studies were excluded, because they did not meet our inclusion criteria (par. 2.2), leaving fifteen articles for this systematic review. Eight additional studies were found by searching the reference lists of selected articles, reviews and systematic reviews on the topic. Hence, a total 23 articles were ultimately selected: twelve on EIBI, ten on ESDM and one on both EIBI and ESDM. The selection process is illustrated in Figure 1. Complete lists of all articles extracted from each database are provided in Appendix A, distinguished by database source (sheets 1–4), and specifying the cause for exclusion (sheet 5). 

### 2.5. Meta-Analytical Strategy for Combining p-Values

To quantitatively systematize the literature data, *p*-values from different studies assessing the same putative predictor were combined using Fisher’s method [55]. We chose these statistics because the association between each putative predictor and treatment outcome was tested using different statistical methods across multiple studies (*t*-tests, ANOVAs, Pearson’s correlation, regression analysis). In brief, Fisher’s method combines *p*-values from *k* independent tests of the same null hypothesis (*H*_0_), into one chi-squared (χ^2^) statistics with 2*k* degrees of freedom, providing a single combined *p*-value [55], as follows:$$X_{2k}^{2}\sim -2\sum_{i=1}^{k}\log\left(p_{i}\right)$$

To perform this meta-analytic procedure, *p*-values were recorded or extrapolated from each study. When more than one association between a putative predictor and an outcome variable was reported in the same study, the smallest *p*-value was chosen. When not explicitly reported, *p*-values were calculated from the available test statistics using GraphPad QuickCalcs Website [https://www.graphpad.com/quickcalcs/pvalue1.cfm (accessed on 4 October 2022)]. Fisher’s method was performed in R version 4.1.2 [56], using the “fisher” function of the “poolr” package [57].

## 3. Results

Overall, twenty-three articles were deemed eligible for inclusion in the systematic review: twelve on EIBI, nine on ESDM, and one on both EIBI and ESDM.

### 3.1. Early Intensive Behavioral Interventions

Thirteen publications reporting child’s predictors of EIBI outcome were selected in this section. These studies include six case-control trials [11,13,26,48,58,59], five single-group pre–post-treatment studies [15,48,59,60,61], and two randomized controlled trials (RCTs) [16,23]. One publication [62] is a two-year follow-up study of the same sample previously reported by Remington et al. [26]; therefore, these two publications will be counted and presented as a single study. Two publications [63,64] identified through database search were excluded because they merged the EIBI group with the comparison group in their results. These studies regarded the same sample as Zachor and Ben-Itzchak [58], which was included instead because the Authors analyzed and reported the experimental and the control groups separately. One study [37] was excluded, because it reports on the data collected from sixteen different individual publications, seven of which are included in the present review. Twenty-four articles on predictors of EIBI outcome, including three follow-up studies (twelve identified through database search and ten through systematic reviews and/or other reference lists) were excluded and will not be discussed because the age range of children at treatment start was over 48 months; however, these studies are listed for consultation in Appendix A).

#### 3.1.1. Sample Characteristics and Patient Selection Criteria

Sample characteristics are summarized in Table 1. Overall, the studies included in this section comprised 382 children with Autism Spectrum Disorder aged 12–48 months at intake, including 38 (10%) females (three studies [26,58,65] did not report the gender of their experimental sample). Eight studies had a comparison group, comprising 220 children with ASD aged 12–42 months, including 33 (15%) females (but two studies [26,58] did not report the gender of their comparison group), and 58 typically developing children aged 18–59 months (gender not reported) [61]. In many studies, it was not possible to establish the age at which treatment actually began with any certainty, only the age at which the child was first referred or diagnosed (e.g., [13,26]).

Selected studies recruited patients diagnosed with ASD according to DSM-IV criteria, except for three studies, which used DSM-III [11], DSM-5 [23] and ICD-10 [15] criteria instead. Standardized instruments used to confirm the clinical diagnosis include the Autism Diagnostic Instrument-Revised (ADI-R) [66], in seven studies [13,15,26,48,58,59,65], the Autism Diagnostic Observation Schedule (ADOS) [67], in five studies [26,58,59,60,65], and the ADOS-2 [68], in one study [23]. Smith and colleagues [16] report that diagnosis for their participants was made by licensed psychologists independently of the study.

Main patient exclusion criteria in these studies were severe medical conditions [13,15,16,23,26,58,65]; genetic syndromes [23,59,60]; neurological disorders [48,59]; significant hearing [23,60], vision, physical or motor impairment, or children not yet walking [23]. Three studies [13,23,48] excluded children with an IQ < 35. Smith and Colleagues [16] did not include children with an IQ below 35 or above 75, while Lovaas [11] excluded children whose mental age was ≤11 months at a chronological age (CA) of 30 months. Four studies reported excluding children based on their CA at referral. Lovaas [11] excluded children whose CA was over 40 months if they were mute, or over 46 months if they were echolalic. Remington and Colleagues [26] included only children between 30 and 42 months old; Sallows and Graupner [48] between 24 and 42 months; Smith et al. [16] between 18 months and 42 months of age. One study [61] did not report any patient inclusion/exclusion criteria.

#### 3.1.2. Treatment

Treatment characteristics are summarized in Table 2. All studies provided individualized EIBI. Six studies [11,13,15,23,48] employed the EIBI developed and manualized by Lovaas and Colleagues [12] at UCLA; three studies [58,60,61] provided EIBI according to Leaf and McEachin [69] and Maurice and Colleagues [70]; two studies [26,65] report to have provided ABA-based intervention using discrete trial teaching and other behavioral techniques. Finally, one study [59] reported providing a center-based ABA program.

Children with ASD in the comparison group received Eclectic Intervention [58,65], ESDM [23], low-intensity EIBI [11], parent-delivered EIBI [16,48], or treatment as usual (TAU) [13,26], also defined “community therapy” in some studies. Lovaas [11] also had a second comparison group, consisting of children studied by a different research group [71]. In one case, [15] the comparison group was composed of parent-commissioned EIBI (i.e., staff was hired and managed by parents themselves, as opposed to university-based EIBI where treatment personnel were provided by the University). Ten out of the 23 children included in Remington et al. [26] intervention group also received parent-commissioned EIBI, but they were considered part of the EIBI group, together with children receiving clinic-delivered EIBI.

The duration of EIBI varied considerably between studies, ranging from one to four years or more. Mean treatment duration was 22 months. Intensity also varied substantially, from a minimum of 12 h/week up to 40 h/week or more. On average, children received 28 h/week. Identifying the intensity and duration of treatment received by children in the comparison groups was more difficult, as the Authors did not always clearly report this information (e.g., [16,26,65]). Nevertheless, children in the comparison group received at least 19 h/week of treatment for approximately 19 months.

#### 3.1.3. Measures

Outcome measures included cognitive abilities, autism symptoms severity, adaptive behaviors, language and communication abilities, social skills, and, in some cases, school placement, motor skills, imitation and joint attention. The measures used to assess IQ include the Bayley Scales of Infant Development (BSID) [72] in eight studies [11,13,15,16,26,48,59,65], the Stanford–Binet Intelligence Scale [73,74] in five studies [11,16,26,59,65], the Wechsler Preschool and Primary Scale of Intelligence–Revised (WPPSI-R) [75] or the Wechsler Intelligence Scale for Children—Revised (WISC-R) [76] in four studies [11,13,15,48], and the Mullen Scales of Early Learning (MSEL) [77] in three studies [23,58,60]. Lovaas [11] also used the Cattell Infant Intelligence Scale [78], the Gesell Infant Development Scale [79], and the Vineland Social Maturity Scale [80] to assess the mental age of some children in their sample. To assess visual-reception skills, five studies [11,13,15,16,48] used the Merrill–Palmer Scale of Mental Tests [81]. Autism symptoms severity was assessed with ADI-R in seven studies [13,15,26,48,58,60,63] and with ADOS in five studies [23,58,59,60,65]. Remington et al. [26] also administered the Autism Screening Questionnaire (ASQ, [82]). Five studies [13,15,16,26,48] assessed receptive and expressive language using the Reynell Developmental Language Scales [83], while Rogers and Colleagues [23] administered the MacArthur–Bates Communicative Developmental Inventories [84]. Eight studies [13,15,16,23,26,48,58,59,60,65] assessed adaptive behaviors with the Vineland Adaptive Behavior Scales (VABS) [85,86]. In some studies, direct-observation tools were used to assess children’s skills, including the Early Learning Measures (ELM) [87], the Early Social Communication Scale (ESCS) [88], and the Early Skills Assessment Tool (ESAT) [89], respectively [16,26,61]. Ben-Itzchak et al. [59] also used developmental-behavioral scales to measure imitation, receptive and expressive language, and restricted and stereotyped behaviors. Remington et al. [26] assessed child behaviors with the Positive Social subscale of the Nisonger Child Behavior Rating Form [90] and the parent-report version of the Developmental Behavior Checklist [91]. Finally, the Child Behavior Checklist (CBCL) [92] was applied in two studies to assess children’s behaviors [16,48].

#### 3.1.4. Predictors of EIBI Treatment Outcome

Pre-treatment characteristics associated with response to EIBI are listed in Table 3. It is important to point out that the results of Hayward et al. [15] and of Sallows and Graupner [48] refer to their entire sample, i.e., to the intervention and control group combined. However, while in Hayward et al. [15], the control group received EIBI commissioned by parents and delivered by trained therapists, in Sallows and Graupner [48], the comparison group received EIBI delivered directly by parents. 

*Cognitive abilities*. Cognitive abilities at baseline represent the most-studied predictor of EIBI outcome in young children. Seven studies reported the association between pre-treatment IQ and EIBI treatment outcome, whereas four studies failed to find a correlation (Table 3). Lovaas [11] found that children with the most favorable outcome (i.e., children who achieved normal educational and intellectual functioning) had a higher IQ and mental age at pre-intervention. In subsequent studies, children with pretreatment IQ or DQ ≥ 70 showed significantly greater improvements in receptive language [59], as well as Communication, Daily living and Socialization VABS sub-domains scores [60]; performed better in ADOS scores both pre- and post-intervention (however, this result was also found in the comparison group) [65] and had a better outcome in terms of improved IQ [26,48], possibly predicting the maintenance of improvement at two-year follow-up [62]. In contrast to these seven positive results, four studies [13,16,23,61] found no association between pre-treatment IQ and EIBI intervention outcome, although one of these studies reported non-significantly higher pre-treatment cognitive scores in children categorized as High/Medium Responders, as compared to Low Responders [61]. Counterintuitively, in two studies, children with IQ < 70 at baseline showed a significantly greater improvement in imitation skills [59] and in MSEL scores, especially Fine Motor and Receptive Language score [60], compared to children with higher pretreatment IQ. Importantly, Hayward and Colleagues [15] found that pre-treatment visuo-spatial IQ correlated not only with post-treatment visuo-spatial IQ, but also with the magnitude of improvement in global IQ, language abilities (both receptive and expressive) and adaptive behaviors at the end of EIBI. 

*Chronological age at intake*. Four studies explored the predictive role of age at treatment onset [11,15,26,61], but only one found younger chronological age at intake to be associated with better EIBI outcome. Specifically, MacDonald and Colleagues [61] found that children under 29 months of age were more likely to be classified as high responders and improved more than their older peers in terms of joint attention, cognitive abilities and play skills. However, the lack of a comparison group receiving another type of treatment does not allow to conclude with any certainty whether this result is specific to EIBI. In the other three studies, age at intake did not predict treatment outcome [11,15,26].

*Severity of autism symptoms*. Results are very mixed, as only one out of the three studies significantly supports a correlation between milder autism symptoms and better outcome and/or longer maintenance of improvement after EIBI (Table 3). In fact, Zachor and Ben-Itchak [58] report that children with milder severity symptoms (in both the EIBI and Eclectic group) showed greater gains in adaptive skills (i.e., VABS Daily Living, Communication and Socialization), cognitive and language abilities. Instead, Remington and Colleagues [26] found that children with the best outcome showed more severe, not milder autistic symptoms, as reported by their parents. However, in their 2-year follow-up study [62] the Authors found that children who maintained the positive effects of EIBI displayed a non-significant (*p* = 0.051) trend toward less severe symptoms of autism, as measured by the ADI-R, upon treatment start. Finally, Rogers and Colleagues [23] did not find any significant association between autism severity and EIBI outcome. 

*Language skills*. In general, studies report that pretreatment language skills were primarily correlated with post-treatment language skills [15]. A broader improvement involving additional functions was described by two out of four studies, reporting a correlation between receptive language and EIBI outcome, with a third study displaying a non-significant trend in this direction (Table 3). Smith et al. [16] reported that language skills at entry were positively correlated with language skills and adaptive behaviors after two years of EIBI. Sallows and Graupner [48] found that receptive language predicted later IQ, social and language skills when considered together with other pre-treatment variables such as imitation, IQ, social interest and communication abilities. Cohen and Colleagues [13] reported that children with the most favorable outcome (i.e., children who scored on the average range on all outcome measures) showed a trend toward a slightly better receptive language at intake, although this finding was not significant. No association was found regarding expressive language [13]. Finally, Rogers and Colleagues [23] did not detect any significant association between pre-treatment language abilities and treatment outcome. 

*Communication skills*. Two out of three studies found more developed communication skills associated with better outcome (Table 3). Remington et al. [26] found that children who benefited the most from treatment had higher communication skills at entry. Sallows and Graupner [48] found that communication abilities predicted later IQ, social and language skills, together with other pre-treatment variables (Table 3). However, Ben-Itzchak and Zachor [59] found no differences in the outcome of children who started treatment with higher vs. lower communication abilities, as measured by the ADOS. Unfortunately, these three studies each used a different tool to assess communication skills (Table 3), and this may have contributed to their discordant results.

*Social skills.* Three studies assessed social skills and found them associated with a better outcome after EIBI treatment (Table 3). Ben-Itzchak and Zachor [59] found that children with higher pre-treatment social skills showed greater improvement in receptive language and a trend toward slightly higher improvement in expressive language. Sallows and Graupner [48] found that social skills predicted post-treatment IQ, language and social skills, together with other variables, as outlined above. Finally, VABS Social Skills scores are part of a panel of variables predictive of EIBI response, measured as IQ change [26], whereas, in the same sample, ADI-R social skill scores predict persistent benefits two years after the end of treatment [62] (Table 3).

*Adaptive behaviors*. Four studies addressed adaptive behaviors, yielding mixed results (Table 3). Sallows and Graupner [48] found that VABS Daily Living Skills was one of several variables, such as imitation, receptive language and communication skills, that best predicted post-treatment IQ, language and social skills. In another study [26], children with the best outcome showed better pretreatment adaptive behaviors, as measured by the VABS, but also greater problem behaviors, as measured by the Developmental Behavior Checklist. Adaptive behaviors were not found to predict EIBI outcome in two other studies [13,15].

*Imitation skills*. Only one study, by Sallows and Graupner [48], investigated and confirmed that verbal and nonverbal imitation strongly predicted post-treatment IQ, social and language skills (Table 3). 

*Joint Attention.* MacDonald and Colleagues [61] reported that High/Medium responders to EIBI had higher tendency to initiate joint attention, but this result did not reach statistical significance.

### 3.2. Early Start Denver Model

Eleven studies on ESDM reporting predictors of outcome were selected. Four articles focused on one-group pre–post-test studies [51,93,94,95] and one was an observational retrospective study [96]. Three studies were randomized controlled trials (RCTs) [22,23,97]. Three studies were case–control trials [98,99,100]. Latrèche and Colleagues [99] conducted both a cross-sectional and a longitudinal analysis, comparing children with ASD with typically developing children and children with ASD receiving ESDM vs. TAU, respectively. Eight additional studies were not included, since some children were older than 48 months at treatment start, but the sample characteristics, intervention strategies and outcome of these studies are summarized for consultation in Appendix A.

#### 3.2.1. Sample Characteristics and Patient Selection Criteria

Overall, these eleven studies investigated predictors of ESDM outcome in 468 children, including 107 (22.8%) females, aged 12–48 months at intake based on our study-selection criteria. A summary of sample characteristics can be found in Table 4. Four studies [22,23,99,100] had a control group, consisting of 206 children with ASD, including 41 (19.9%) females, aged 12–48 months at intake. Latrèche and Colleagues [99] also included a second comparison group consisting of 16 typically developing children (females *n* = 4, 25.0%). One study [98] compared outcomes of younger children (18–48 months) with 28 older children with ASD aged 48–62 months at intake, both receiving ESDM. 

Four studies included patients with ASD diagnosed according to DSM-5 criteria [23,95,96,97,98,99,100], whereas one study used DSM-IV criteria for Autistic Disorder or PDD-NOS [22]. All studies administered ADOS or ADOS-2 to confirm the diagnosis. 

Main patient exclusion criteria in selected studies were severe medical conditions other than ASD [22,23,95,98,100], neurological disorders [51,95,96,100], genetic syndromes [51,95,96] and significant vision, hearing, motor, or physical impairment [22,23,51,96,98,100]. Two studies [22,23] excluded children with a Developmental Quotient (DQ) below 35 on MSEL; children born at a gestational age of less than 34 months; and children not yet walking. One study [97] reported no exclusion criteria based on child behaviors or cognitive abilities. Finally, three studies [93,94,99] did not specify exclusion/inclusion criteria. 

#### 3.2.2. Treatment

ESDM treatment characteristics are summarized in Table 5. All eleven studies implemented ESDM according to the Rogers and Dawson [19] manual. Seven studies [22,23,94,95,96,99,100] delivered individualized ESDM sessions, three [93,97,98] delivered group-setting ESDM, while Contaldo and Colleagues [51] provided two hours of individualized and two hours of group ESDM sessions per week. Children received, on average, 13 h/week of ESDM (range: 3–20) for an overall mean duration of 15 months (range: 10–24). Children in comparison groups received community therapy (CT) [22,99], also defined as “treatment as usual” (TAU) in some studies, or EIBI [23], for an average of 14 h/week (range: 3.4–20) for approximately 18 months (range 12–24). Vivanti et al. [98] provided ESDM to children in the older group with the same intensity (20 h/week) and for the same duration (12 months) as children in the younger group. Finally, children in the Wang and Colleagues [100] comparison group were on a waiting list to ESDM.

#### 3.2.3. Measures

Outcome measures included mainly cognitive abilities, autism severity, adaptive behaviors, language and communication abilities, social skills, joint attention, and imitation. Cognitive abilities were assessed using the MSEL in all studies except three [51,96,100], which administered, respectively, the Bayley Scales of Infant Development-Third Edition (Bayley-III) [101] and the Wechsler Preschool and Primary Scale of Intelligence-Third Edition (WPPSI-III) [102], the Gesell Developmental Scale (GDS) [103] and the Griffith Mental Development Scales (GMDS) [104], which was also administered by Vivanti et al. [97]. One study [95] assessed children’s cognition, motor and adaptive skills through the Psychoeducational Profile-Third Edition (PEP—3) [105] ADOS-2 or ADOS were used by all studies to assess autism severity. One study [51] also employed the Childhood Autism Rating Scale (CARS) [106]. Adaptive behaviors were assessed with VABS-II in all studies except one [96], while Zitter and Colleagues [94] administered VABS-3 [107]. Contaldo et al. [51] administered the Italian version of the MB-CDI (“Il Primo Vocabolario del Bambino”) [108] to assess language skills, and the ESDM Curriculum Checklist [19], a direct-observation tool used to evaluate children’s skills in all developmental domains. One study [93] employed the Language Environment Analysis System (LENA), a wearable audio-recorder used to capture and quantify child vocalization and language-learning environment (LENA Research Foundation) [109]. Finally, Latrèche and Colleagues [99] used an eye-tracking paradigm to measure children’s attention to faces.

#### 3.2.4. Predictors of ESDM Treatment Outcome

Pre-treatment characteristics associated with response to ESDM are listed in Table 6.

*Cognitive abilities*. Five studies found an association between cognitive abilities and several outcome measures [22,51,93,95,96], in contrast to two negative studies [23,94]. In particular, Contaldo and Colleagues [51] found that a higher developmental age at entry was associated with faster gains in “Socialization”, and “Cognition and Play” ESDM-checklist domains, as well as the rate of learning (operationalized as the number of objects acquired in one months by each child). Rogers and Colleagues [22] found that children with a higher DQ at baseline had lower autistic scores on the ADOS at the end of treatment. Two studies [93,95] found that DQ at baseline predicted DQ at the end of treatment. Surprisingly, one study [96] found that children with a DQ below 75 at baseline showed greater post-treatment improvement in cognitive and language scores compared to their peers, whose DQ was ≥75. Two studies [23,94] did not find any association between cognitive abilities and any outcome measure.

*Chronological age at intake*. Five studies addressed the possible association between age at treatment onset and final outcome. Pre-treatment chronological age was found to predict ESDM outcome in four of these five studies. Devescovi and Colleagues [96] found that entering ESDM before 27 months predicted greater improvements in autistic symptoms severity. Vivanti and Colleagues [98] compared younger children (18–48 months) with older children (48–62 months) receiving ESDM and found that younger children reached significantly larger gains on verbal DQ after one year of treatment, and that this result was moderated by initial language skills. Similarly, the same group also found that a younger age predicted verbal DQ, regardless of other factors [97]. Counterintuitively, Zitter et al. [94] found that age at intake was positively correlated with child-learning response; that is, older children responded more quickly to ESDM. Finally, no association between chronological age and any outcome measure was reported by Contaldo and Colleagues [51].

*Severity of autism symptoms*. Six studies investigated the possible association between ASD symptom severity and treatment outcome [22,23,51,94,95,100]. All but one [51] yielded negative results. In the only positive study, milder autism severity predicted greater gains in Socialization, Cognition, Play, and Motor ESDM-checklist domains, as well as in the rate of learning (i.e., number of learning objectives acquired by each child in one month) [51]. 

*Language skills*. Three out of four studies support receptive language and non-verbal communication as predictive of outcome after ESDM. Sulek and Colleagues [93] found that children vocalization ratio (i.e., a measure of speech-related sounds compared to non-speech sounds, such as vegetative sounds) was predictive of post-treatment DQ, together with pre-treatment DQ. Similarly, Godel and Colleagues [95] found that Expressive and Receptive Language predicted DQ and rate of DQ change at the end of treatment. Contaldo and Colleagues [51] found that receptive language, but not word production, was significantly associated with gains in Socialization, Cognition and Play, and Motor ESDM-checklist domains, as well as the rate of learning. Non-verbal communication, notably first communicative gestures repertoire and action with objects, were also associated with greater gains in the Communication ESDM-checklist domain [51]. Instead, Rogers and Colleagues [22] found that language abilities did not influence the effect of ESDM.

*Attention to faces*. One study [99] reports that higher levels of attention to faces, operationalized as the percentage of time spent staring at a face measured through an eye-tracking task, is predictive of children showing higher gains in overall DQ and verbal DQ after ESDM.

*Stereotyped and repetitive behaviors.* Three studies investigated whether this factor was associated with ESDM response. One of these [94] reported no significant results, while the other two found that lower repetitive behaviors at baseline predicted improvement in overall DQ [95] and cognitive verbal/preverbal [100] post-treatment.

*Imitation, joint attention, play skills, and adaptive skills*. Imitation [51], joint attention [22], and play skills [22] did not possess significant predictive power on ESDM outcome in single studies involving young children. Better adaptive skills were found to predict improvement in post-treatment DQ in only one study [95], while another was negative [94]. Similarly, studies enrolling older children reported mixed results (Appendix A). 

## 4. Discussion

One of the ultimate aims of autism research is to allow for clinicians to define “which treatment for which child” beforehand, based on clinical predictors and objective biomarkers (genetics, brain imaging, electrophysiology, eye tracking, etc.). This aim not only regards pharmacological therapy [7,8], but also behavioral interventions which, although sharing some common elements, differ significantly in multiple aspects of their methodology. Structured behavioral approaches tend to favor a “teaching” relationship, and employ tasks which preferentially request and strengthen cognitive skills, making broader use of extrinsic motivators [9]; naturalistic approaches employ a “playground-like” relationship, employing activities that leave greater freedom of choice to the child and act as intrinsic motivators, while primarily requiring and strengthening social cognition (eye contact, theory of mind, joint attention, empathy, etc.). Predictably, not all children respond equally well to early intervention approaches, and yet it is at this time, early in life, that it would be most useful to provide targeted treatments, to maximally exploit neural plasticity. Research on pretreatment predictors of greater gains after behavioral interventions is still in its infancy. The available evidence trying to link preferential response to a specific type of treatment with a set of clinical/demographic characteristics is even more incomplete. Nonetheless, the evidence we have collected begins to point in some directions, which can possibly begin to orient clinicians and provide useful hints for future hypothesis-driven studies. 

With this aim in mind, we began to identify first- and second-line predictors, as summarized in Table 7. First, we considered the number of studies addressing each pre-treatment variable in connection with post-treatment outcome. We then quantified the amount of available evidence in favor of each putative predictor, in terms of number of studies reporting a positive association between predictor and outcome, as well as cumulative *p*-value for each obtained predictor, combining all published statistical outcomes from multiple studies using Fisher’s method (Table 7 and Appendix A). First-line predictors are supported by more than 50% of the available studies for each treatment approach, with a cumulative *p*-values in the range of 10^−10^–10^−11^. Second-line predictors appear promising, as they also are supported by at least 50% of the available studies, but have been assessed in fewer articles and/or yield a cumulative *p*-value below the above-mentioned range. Other pre-treatment variables appear, at this stage, to be “Weak or non-predictors”, because they have been found to be associated with outcome in a minority of studies and/or with cumulative *p*-values < 10^−5^. Finally, special caution is required with variables assessed only in one or two studies, as insufficient evidence is currently available (Table 7). 

Applying this stratification framework to studies regarding *EIBI*, the most studied and reliable factor associated with outcome in young autistic children is *IQ/DQ at intake*, since seven out of eleven studies support its predictive power, reaching an impressive combined *p*-value (Table 7). Interestingly, visuo-spatial IQ can especially be developed in EIBI responders [8]. Promising second-line predictors of better outcome after EIBI, requiring more studies to conclusively confirm and quantify their predictive power, include better receptive language abilities, communication skills, and social skills. Variables unlikely to be associated with EIBI outcome surprisingly include younger age at intake, and milder severity of autistic symptoms. Adaptive behaviors also yield very mixed results, which may reflect that this is a complex construct engaging multiple underlying skills. No conclusions can be drawn at this time about imitation and joint attention, which have each been the object of a single study to date.

For *ESDM*, the broader construct of pretreatment “*social cognition*” appears to predict a positive response in five out of six studies assessing communication (verbal and non-verbal) and attention to faces (Table 7). In particular, verbal (receptive and expressive language) and non-verbal (gestures) communication skills were collectively assessed in five studies, one negative [22] and four documenting greater improvements associated with better language skills at intake [51,98], higher expressive and receptive language DQ at the Mullen Scales of Early Learning (MSEL) [95], a broader repertoire of first communicative gestures [51], and greater intentional communication in the form of more speech-related vs. non-speech related vocalizations [93], albeit not necessarily full word production [51]. Meanwhile, another key feature in social cognition, i.e., attention to faces assessed by eye-tracking, was also predictive of better response to ESDM in another study [99]. Promising, but more mixed results, concerned age at intake, pretreatment IQ/DQ, and stereotyped/repetitive behaviors. Five studies investigated the predictive role of age at intake: three found that younger children made the biggest progress [96,97,98], one study found that older children achieved greater improvements [94], while one study was negative [51] (Table 6). Six studies investigated the predictive power of DQ at the beginning of treatment over response to ESDM, with four studies finding an association between greater post-treatment response and higher pre-treatment DQ [22,51,93,95], one study finding lower DQ predictive of greater post-treatment gains [96], and two studies reporting no association [23,94]. Interestingly, most studies tend to exclude a predictive role for the severity of autism symptoms prior to ESDM, which, despite being addressed by six studies, only reaches a cumulative *p*-value of 0.032 (Table 7). More research is needed, especially research focused on younger children, to draw firm conclusions on adaptive behaviors, imitation, joint attention, and play skills, each mostly not supported by single studies (Table 6), but with some positive results in research involving older children (Appendix A). 

Predictors of positive response to EIBI and ESDM partially overlap, but also display some interesting differences. As expected, IQ/DQ was the most frequently reported variable associated with response to EIBI, but an association was also found in ESDM studies, although this was not as strong (Table 7). This difference is also present in studies involving older children, whereby IQ/DQ predicts better outcome to EIBI and ESDM in 11/12 (91.7%) and in 4/7 (57.1%) studies, respectively (Appendix A). Conversely, response to ESDM was often associated with a set of variables falling within the realm of “social cognition”, including more speech-related sounds (i.e., greater intention to communicate), better receptive and expressive language, and greater attention to faces (Table 6 and Table 7). Some of these “social” variables also partly predict response to EIBI, but not quite as convincingly (Table 7). On the one hand, the overlap is not surprising, because both EIBI and ESDM propose tasks whose learning is influenced by child IQ/DQ, and both require interpersonal interactions between child and therapist. On the other hand, these results are collectively beginning to delineate an important difference: IQ/DQ and social cognition may represent *preferential* predictors of response to EIBI vs. ESDM, respectively, because these functions are the most required by each approach and may benefit the most from each approach. More specifically, children who have better cognitive functions, systemizing skills, and visuo-spatial IQ may benefit more from structured approaches that largely employ these functions, whereas children endowed with greater social motivation and with milder deficits in social cognition and communication may benefit more from approaches that use play, child’s initiative and fun interactions as a primary channel for stimulation. At the same time, EIBI and ESDM may catalyze global development and broader adaptive skills by primarily strengthening cognitive and social functions, respectively. Nonetheless, given the available evidence, this statement must be viewed more as a rationale hypothesis with some promising initial support than as a firm conclusion, which will require additional research.

There is some evidence that children may benefit more from starting interventions at an earlier age, especially for ESDM (Table 6). However, results are mixed and perhaps even disappointing for EIBI (Table 4 and Table 7). Some researchers [51,98] suggest that this might be due to the very narrow age-range of the children enrolled in most studies. However, even studies with a wider age-range have reported variable results, with only 10/19 (52.6%) and 1/6 (16.6%) studies finding that younger children achieve better results with EIBI and ESDM, respectively (see Appendix A). Efficacy also in slightly older children suggests that the critical period of maximum plasticity, allowing for a satisfactory response to any type of early intensive intervention for some children could conceivably last longer than the narrow time-window adopted in this review (i.e., for some children, even starting treatment at 4–5 years of age may foster a positive response). At the same time, this lack of consistent benefits in children whose treatment was started at a very early age clearly shows that there are other variables that can override the effect of age on neuroplastic responses to environmental stimulation. One of these variables could conceivably be represented by rare [7] and common [110] genetic variants that negatively modulate dendritic spine formation and synaptic functions, including LTP and LTD. In parallel, similarly mixed results were obtained with autism severity in both EIBI and ESDM, investigated in three and six studies, respectively (Table 3 and Table 6). Milder symptoms at treatment onset were only associated with better outcome in 1/3 (33.3%) EIBI studies and in 1/6 (16.6%) ESDM study (Table 7). Studies recruiting older children yielded positive results in 4/9 (44.4%) for EIBI and 4/6 (66.7%) studies for ESDM (Appendix A). This apparently greater predictive power for ESDM, if not a chance finding, may indicate that the severity of autistic symptoms is less relevant and predictive when treatment is started early, but may begin to matter more as treatment is started in children 4 years or older. 

The methodological rigor of the studies included in this review was assessed through the Critical Review Form for Quantitative Studies [111], which highlighted several limitations, such as a small sample size yielding interesting trends, which often did not reach statistical significance [13,16], lack of a comparison group receiving another treatment, and lack of randomized assignment to intervention. Only two out of thirteen studies on EIBI and three out of nine studies on ESDM were fully RCTs. In some cases, the treatment protocol was not described in sufficient detail, including its intensity and duration. Studies did not always clearly report the time elapsed between the first assessment and the start of treatment, making it difficult to define the actual age of children at the beginning of the intervention. In some studies, the clinical and psychodiagnostic assessment was made a few months after the start of treatment [99]. Different instruments were used to measure the same variable, such as IQ and language skills, sometimes even within the same study, thus preventing a true comparison of the results. The use of objective measures such as gaze parameters obtained using eye-tracking technologies has been very limited, at least in young children [99]. Some important functions, such as imitation and joint attention, were investigated in very few studies. This is especially surprising, since it has been suggested that these skills may be predictors of positive response to ESDM, given its focus on social and communication skills [112]. This insufficient number of studies does not stem from a bias introduced by our inclusion criteria, because very few studies involving also older children have investigated these critical functions [28,45,112]. On the contrary, it would be advisable to shift away from broader constructs, such as IQ and language skills, and toward more proximal predictors of outcome, such as spontaneous imitation, vocalizations, and social interaction [47]. Furthermore, investigators often find what they are searching for: as an example, proportionally fewer ESDM studies investigated IQ/DQ as predictor of treatment outcome compared to EIBI studies (7/11 studies = 63.6% for ESDM, as compared to 11/13 studies = 84.6% of EIBI studies). Several studies have found that “overall pre-treatment functioning” or “initial learning rate” are associated with later outcome [113,114,115,116,117], and this may lead to circular reasoning: children who are more likely to learn because they are skillful will learn more and sooner from interventions [47]. However, many other factors and confounding variables may influence the initial rate of learning, for example, the degree of response to reinforcers used during treatment [116]. Moreover, early intensive interventions should act as catalyzers of functions which, at a given time, are observed to be underdeveloped in a child: placing this process into the framework of a mere “learning” paradigm may well be oversimplifying the complexities of motivation, emotion and relationship, which are at the core of autistic deficits. This point is often neglected in studies whose outcome measures are exclusively focused on cognitive functioning and DQ. Hopefully, future research will take these limitations into account, to reach broader and more definitive conclusions [24,47,118].

## 5. Limitations and Strengths

In many countries, the age at first diagnosis is beginning to significantly decrease, thanks to the more efficient and targeted health policies, increasing social engagement, growing political attention on autism, and larger resources being invested in ASD compared to the past. For this reason, we decided to focus the present review on young children, selecting studies whose sample was limited to the 12–48-month age range at the start of treatment. On the one hand, this stringent approach is a strength of our systematic review and metanalysis, because children do change with age and early infancy is when children should receive their first diagnosis and intensive intervention. Including studies also enrolling children who were older than 48 months at treatment onset can be expected to decrease the sensitivity and specificity of behavioral predictors of treatment response, because of heterogeneity in the timing of children’s developmental trajectory. Nonetheless, these studies are provided to interested readers in Appendix A. On the other hand, the exclusion of studies focused on parent-delivered interventions may be viewed as a limitation. We made this choice to avoid mixing parent- and therapist-delivered interventions, which are intrinsically different, and to prevent parental variables, such as parental stress, from introducing a bias in our assessment of treatment efficacy [119]. Furthermore, we only analyzed the intrinsic characteristics of the child as predictors of treatment outcome, under the assumption that treatments would be delivered by equally expert personnel using manualized approaches. Treatment-related factors, such as treatment intensity [37], influence response in real life and have not been considered here. Finally, the stratification strategy that we adopted has its own strengths and limitations. We attempted to “weigh” the evidence in favor of or against single predictors, by considering both the quantity of available studies and the effect size of the association between each predictor and treatment outcome. Given the limited number of articles selected in this review, the different possible strategies that are available to combine *p*-values, and the fact that our stratification criteria appear well-justified but also somewhat subjective, our conclusions should be viewed as preliminary rather than definitive.

## 6. Conclusions

A number of studies have now proven the effectiveness of comprehensive behavioral interventions such as EIBI and ESDM. Predictably, great individual variability has been observed in response to treatment, with some children showing considerably larger gains than others. The aim of this systematic review and meta-analysis was to begin identifying factors associated with positive response to early intervention and possibly treatment-specific predictors for EIBI and ESDM, which may help to maximize the clinical efficacy of intervention strategies. Our systematic review and combined *p*-values indicate that at a very young age, cognitive skills and developmental quotient appear to be most predictive of greater gains with EIBI. Instead, a set of variables pertaining to social cognition and communication appear to be most predictive of response to ESDM. More research focused on young children aged 12–48 mo and possibly devoid of the methodological limitations present in many studies published to date, will be necessary to draw more firm conclusions on these and on other promising variables that have been presented and discussed here (Table 7). Nonetheless, the results of this systematic review begin to shed some light on the factors associated with preferential response to EIBI and ESDM. Despite our many caveats and the need to always consider the broader clinical context of each patient, this information may provide clinicians with some useful clues when personalizing intervention strategies for young children newly diagnosed with ASD.

## Figures and Tables

**Figure 1 brainsci-12-01499-f001:**
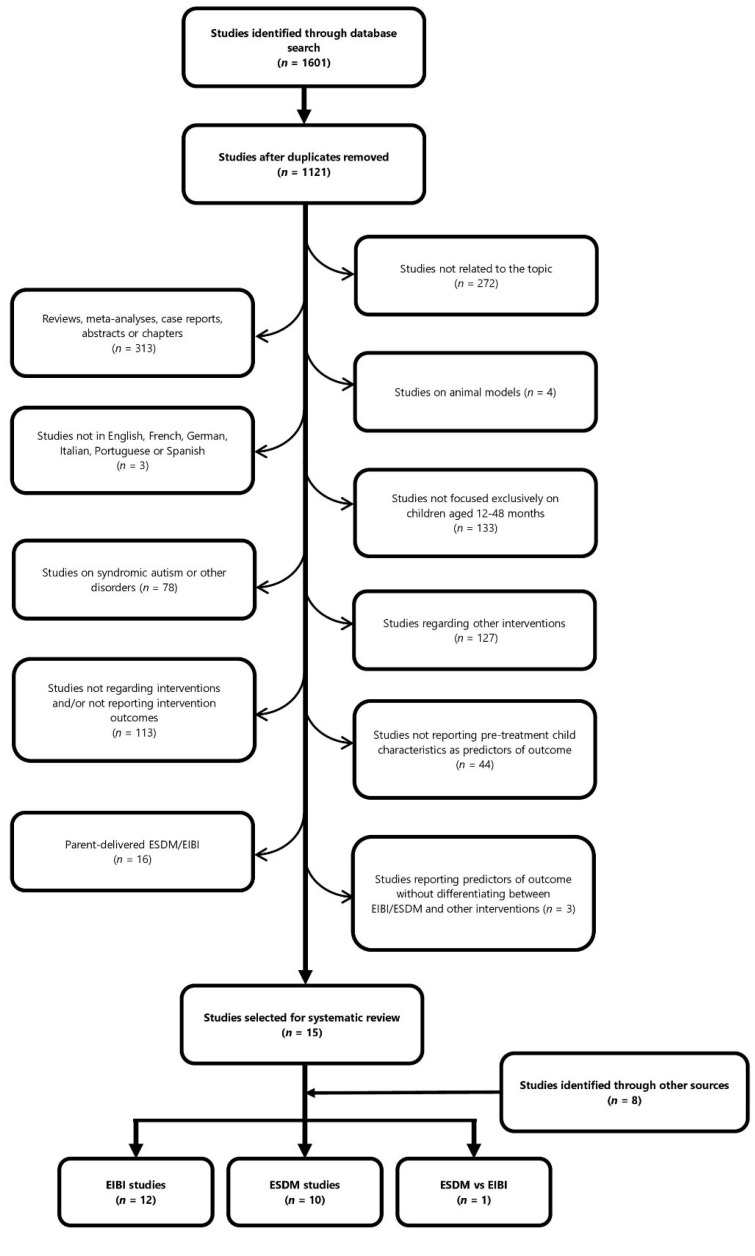
Study flow chart. EIBI: Early Intensive Behavioral Intervention; ESDM: Early Start Denver Model.

**Table 1 brainsci-12-01499-t001:** Summary of EIBI studies: sample characteristics.

Study	Cases				Controls			
	N (M:F)	Age at Intake in Months (Mean)	Diagnosis	Exclusion Criteria	Control Intervention	N (M:F)	Age at Intake in Months (Mean)	Diagnosis
[59] Ben-Itzchak and Zachor, 2007	29 (25:4)	20–32 (27)	DSM-IV; ADOS; ADI-R	Genetic syndromes Seizure disorder	---	---	---	---
[60] Ben-Itzchak et al., 2014	46 (39:7)	17–33 (25.5)	DSM-IV; ADOS	Genetic syndromes Hearing impairment	---	---	---	---
[13] Cohen et al., 2006	21 (18:3)	20–41 (30)	ADI-R	IQ < 35 Severe medical conditions	TAU	21 (17:4)	20–41 (33)	ASD
[15] Hayward et al., 2009	23 (19:4)	24–42 (36)	ICD-10; ADI-R	Severe medical conditions	Parent-commissioned EIBI	21 (15:6)	24–42 (34)	ASD
[11] Lovaas, 1987	19 (16:3)	<46 (35)	DSM-III	Age at intake > 40 mo. if non-verbal or >46 mo. if echolalic MA ≤ 11 mo. at CA of 30 mo.	C1: low intensity EIBI; C2: none	C1: 19 (11:8); C2: 21 (n.r.)	C1: <42 (41); C2: <42 (n.r.)	C1: ASD; C2:ASD
[61] MacDonald et al., 2014	83 (n.r.)	17–48 (n.r.)	DSM-IV	n.r.	None	58 (n.r.)	18–59	TD
[26] Remington et al., 2007; [62] Kovshoff et al., 2011	23 (n.r.)	30–42 (36)	DSM-IV; ADI-R	Age at intake < 30 and >42 mo. Severe medical conditions	TAU	21 (n.r.)	30–42 (38)	ASD
[23] Rogers et al., 2021	45 (34:11)	12–30 (23)	DSM-5; ADOS-2	Severe medical/genetic conditions; Significant vision, hearing, motor, or physical problems; DQ < 35; Children not yet walking	ESDM	42 (32:10)	12–30 (24)	ASD
[48] Sallows and Graupner, 2005	13 (11:2)	24–42 (35)	DSM-IV; ADI-R	DQ < 35; Age at intake < 24 and >42 mo. Neurological disorders	P-EIBI	10 (8:2)	24–42 (37)	ASD
[16] Smith et al., 2000	15 (12:3)	18–42 (36)	n.r.	Age at intake < 18 and >42 mo. IQ < 35 and >75 Severe medical conditions	P-EIBI	13 (11:2)	18–42 (36)	ASD
[58] Zachor and Ben-Itzchak, 2010	45 (n.r.)	17–35 (25)	DSM-IV; ADOS; ADI-R	Severe medical conditions	Eclectic	33 (n.r.)	15–33 (26)	ASD
[65] Zachor et al., 2007	20 (19:1)	22–34 (28)	DSM-IV; ADOS; ADI-R	Severe medical conditions	Eclectic	19 (18:1)	23–33 (29)	ASD

ADI-R: Autism Diagnostic Interview-Revised; ADOS: Autism Diagnostic Observation Schedule; ASD: Autism Spectrum Disorder; CA: Chronological Age; DQ: Developmental Quotient; DSM: Diagnostic and Statistical Manual of Mental Disorders; EIBI: Early Intensive Behavioral Intervention; ESDM: Early Start Denver Model; F: Females; ICD: International Classification of Diseases; IQ: Intellectual Quotient; M: Males; MA: Mental Age; n.r.: Not reported; P-EIBI: Parent-Delivered Early Intensive Behavioral Intervention; TAU: Treatment as Usual; TD: Typical Development.

**Table 2 brainsci-12-01499-t002:** Summary of EIBI studies: intervention characteristics.

Study	Country	Study Design	Intervention Type	Setting	Intensity	Duration
[59] Ben-Itzchak and Zachor, 2007	Israel	One group pre-test–post-test	ABA	Autism-specific preschool programs	35 h/week	12 mo
[60] Ben-Itzchak et al., 2014	Israel	One group pre-test–post-test	ABA	Centre-based	20 h/week	24 mo
[13] Cohen et al., 2006	USA	Case–control trial	UCLA EIBI	Home-based	35–40 h/week	36 mo
[15] Hayward et al., 2009	UK	Non-concurrent multiple baseline design	UCLA EIBI	Home-based	37 h/week	12 mo
[11] Lovaas, 1987	USA	Case–control trial	UCLA EIBI	Home/School	≥40 h/week	≥24 mo
[61] MacDonald et al., 2014	USA	Case–control trial	ABA	Home/School	20–30 h/week	12 mo
[26] Remington et al., 2007; [62] Kovshoff et al., 2011	UK	Case–control trial; 2-year follow-up	ABA	Home-based	18–34 h/week (mean = 26)	12 mo
[23] Rogers et al., 2021	USA	RCT	UCLA EIBI	Home/Childcare setting	12 vs. 20 h/week	12 mo
[48] Sallows and Graupner, 2005	USA	Case–control trial	UCLA EIBI	Not reported	38 h/week (gradually decreasing when children entered school)	48 mo
[16] Smith et al., 2000	USA	RCT	UCLA EIBI	Home/Preschool	24 h/week (gradually decreasing after the first year)	24–36 mo (mean = 33)
[58] Zachor and Ben-Itzchak, 2010	Israel	Case–control trial	ABA	Autism-specific preschool programs	20 h/week	12 mo
[65] Zachor et al., 2007	Israel	Case–control trial	ABA	Autism-specific preschool programs	35 h/week	12 mo

ABA: Applied Behavior Analysis; EIBI: Early Intensive Behavioral Intervention; RCT: Randomized Controlled Trial; UCLA: University of California, Los Angeles.

**Table 3 brainsci-12-01499-t003:** Predictors of better outcome after EIBI treatment.

Study	Predictors of Better Outcome	Improved Functions Correlated with Predictors	Non-Predictors
[65] Ben-Itzchak and Zachor, 2007	IQ—Higher IQ—Lower Better social skills	Receptive language and play skills Imitation Receptive language (n.s. trend also for expressive language)	Communication skills (ADOS)
[60] Ben-Itzchak et al., 2014	IQ—Higher IQ—Lower	VABS Communication, Daily living skills and Socialization scores MSEL scores, especially Fine Motor and Receptive Language	None reported
[13] Cohen et al., 2006	None reported		IQ Language skills (n.s. trend for receptive language) Adaptive behaviors
[15] Hayward et al., 2009	Higher visuo-spatial IQ	Total IQ Expressive and receptive language Adaptive behaviors	Chronological age Adaptive behaviors Receptive and expressive language
[11] Lovaas, 1987	Higher IQ/mental age	Intellectual and educational functioning	Chronological age
[61] MacDonald et al., 2014	Younger chronological age	Responding to joint attention Initiating joint attention Cognition Play skills	Cognitive abilities Joint Attention (n.s. trend)
[26] Remington et al., 2007 [62] Kovshoff et al., 2011	Higher IQ/ mental age; VABS scores: higher for Adaptive behaviors, Communication, Social skills; lower for motor skills. More behavioral problems Greater severity of autism symptoms ADI-R social skills	IQ Persistent benefits from EIBI (follow-up two years after the end of treatment)	Chronological Age Milder severity of autism symptoms (trend)
[23] Rogers et al., 2021	None reported	None reported	Autism symptom severity MSEL DQ
[48] Sallows and Graupner, 2005	Imitation (verbal and non verbal) Higher IQ Better receptive language ADI-R communication ADI-R social skills VABS daily living skills	IQ Social skills Language skills	None reported
[16] Smith et al., 2000	Language skills	Language skills Adaptive behavior	IQ
[58] Zachor and Ben-Itzchak, 2010	Milder severity of autism symptoms	Adaptive skills (VABS Daily living, Communication and Socialization) Cognitive level Language abilities	None reported
[65] Zachor et al., 2007	Higher IQ	Lower ADOS scores	None reported

AD: Autistic Disorder; ADI-R: Autism Diagnostic Interview-Revised; ADOS: Autism Diagnostic Observation Schedule; IQ: Intellectual Quotient; MSEL: Mullen Scales of Early Learning; n.s.: non-significant; PDD-NOS: Pervasive Developmental Disorder Not Otherwise Specified; VABS: Vineland Adaptive Behavior Scale.

**Table 4 brainsci-12-01499-t004:** Summary of ESDM studies: sample characteristics.

Study	Cases				Controls			
	N (M:F)	Age at Intake in Months (Mean)	Diagnosis	Exclusion Criteria	Control Intervention	N (M:F)	Age at Intake in Months (Mean)	Diagnosis
[51] Contaldo et al., 2019	32 (26:6)	18–39 (29)	ADOS-2	Genetic syndromes Neurological disorders Significant vision, hearing, motor, or physical impairment.	---	---	---	---
[96] Devescovi et al., 2016	21 (18:3)	20–36 (27)	DSM-5; ADOS-2	Genetic syndromes Neurological disorders Significant vision, hearing, motor, or physical impairment.	---	---	---	---
[95] Godel et al., 2022	55 (48:7)	15–42 (28.7)	DSM-5 ADOS-2	Severe somatic, neurologic or genetic condition that could have affected the validity of behavioral measures (e.g., cerebral palsy, epilepsy, etc.)	---	---	---	---
[99] Latrèche et al., 2021	51 (45:6)	17–48 (34)	ADOS-2	n.r.	C1 = CT; C2 = None	C1: 30 (25:5) C2: 16 (12:4)	C1: 17–48 (34); C2: 17–48 (30)	C1: ASD C2: None
[22] Rogers et al., 2019	55 (41:14)	14–29 (21)	DSM-IV; ADOS-2	Severe medical/genetic conditions DQ < 35 Gestational age < 35 wks Children not yet walking	CT	63 (51:12)	14–29 (21)	ASD
[23] Rogers et al., 2021	42 (32:10)	12–30 (24)	DSM-5; ADOS-2	Severe medical/genetic conditions DQ < 35 Significant vision, hearing, motor, or physical impairment Children not yet walking	EIBI	45 (34:11)	12–30 (23)	ASD
[93] Sulek et al., 2022	99 (70:29)	14–47 (32)	ADOS-2	n.r.	---	---	---	---
[98] Vivanti et al., 2016	32 (26:6)	18–48 (33)	DSM-5; ADOS	Severe medical/genetic conditions Significant vision, hearing, motor, or physical impairment	ESDM	28 (25:3)	48–62 (49.5)	ASD
[97] Vivanti et al., 2019	44 (27:17)	15–32 (26)	DSM-5; ADOS-2	No exclusion criteria based on child behavior or cognition	---	---	---	---
[100] Wang et al., 2022	21 (17:4)	18–36 (21)	DSM-5 ADOS	Neurodevelopmental disorders of known genetic etiology Developmental disorder other than ASD Cerebral palsy Hearing disorder	None (Waitlist for ESDM)	24 (18:6)	18–36 (22)	ASD
[94] Zitter et al., 2021	16 (11:5)	20–39 (29)	ADOS-2	n.r.	---	---	---	---

ADOS: Autism Diagnostic Observation Schedule; ASD: Autism Spectrum Disorder; CT: Community Therapy; DQ: Developmental Quotient; DSM: Diagnostic and Statistical Manual of Mental Disorders; EIBI: Early Intensive Behavioral Intervention; ESDM: Early Start Denver Model; n.r.: not reported.

**Table 5 brainsci-12-01499-t005:** Summary of ESDM studies: intervention characteristics.

Study	Country	Study Design	Setting	Intensity	Duration
[51] Contaldo et al., 2019	Italy	One group pretest-posttest	Community-based (GS)	4 h/week (2 h GS and 2 h 1:1)	8–16 mo (mean 12 mo)
[96] Devescovi et al., 2016	Italy	Retrospective study	Community-based	3 h/week	11–19 mo (mean 15 mo)
[95] Godel et al., 2022	Switzerland	One group pretest-posttest	Center-based	20 h/week	24 mo
[99] Latrèche et al., 2021	Switzerland	Case–control trial	n.r.	20 h/week	24 mo
[22] Rogers et al., 2019	USA	RCT	Home/Preschool/Daycare	20 h/week	24 mo
[23] Rogers et al., 2021	USA	RCT	Home/Daycare	12 vs. 20 h/week	12 mo
[93] Sulek et al., 2022	Australia	One group pretest-posttest	Childcare setting (GS)	~15 h/week	12 mo
[98] Vivanti et al., 2016	Australia	Case–control trial	University-based (GS)	15–25 h/week	12 mo
[97] Vivanti et al., 2019	Australia	RCT	School-based (GS)	15 h/week	10 mo
[100] Wang et al., 2022	China	Case–control trial	Hospital-based	1 h/week	6 mo
[94] Zitter et al., 2021	USA	One group pretest-posttest	Clinic-based	4 h/week	12 mo

DQ: Developmental Quotient; GS: Group-setting ESDM; RCT: Randomized Controlled Trial. 3.2.4. Predictors of ESDM treatment outcome.

**Table 6 brainsci-12-01499-t006:** Predictors of positive outcome after ESDM treatment.

Study	Predictors of Better Outcome	Improved Functions Correlated with Predictors	Non-Predictors
[51] Contaldo et al., 2019	Receptive language Higher DQ Lower autism symptoms severity First communicative gestures repertoire Action with objects	Socialization, cognition, play and motor ESDM-checklist domains Rate of learning Communication ESDM-checklist domain	Age at intake Imitation Word production
[96] Devescovi et al., 2016	Younger age at intake Lower DQ (<75) at entry	Greater improvement in severity of autism symptoms Greater improvement in cognitive and language scores	None reported
[95] Godel et al., 2022	Higher MSEL Composite DQ at entry Higher VABS-II Adaptive Behavior Composite and Communication score Higher Expressive and Receptive Language MSEL DQ Higher Visual Reception MSEL DQ Higher Fine Motricity MSEL DQ Lower stereotyped and repetitive behaviors (ADOS RRB) Early developmental progress (i.e., rate of change) by 6 months of intervention	Rate of DQ change	Symptom severity (ADOS CSS)
[99] Latrèche et al., 2021	Attention to faces	MSEL DQ Verbal DQ	None reported
[22] Rogers et al., 2019	Higher DQ at baseline	Lower ADOS scores	Joint Attention Severity of autism symptoms Play skills Expressive and receptive language skills
[23] Rogers et al., 2021	None		Severity of autism symptoms MSEL DQ
[93] Sulek et al., 2022	MSEL DQ Speech-related vocalization ratio ^1^	MSEL DQ	None reported
[98] Vivanti et al., 2016	Younger age at intake Initial language	Verbal DQ	None reported
[97] Vivanti et al., 2019	Younger age at intake	Verbal DQ	
[100] Wang et al., 2022	Less stereotyped and repetitive behaviors (ADOS RRB)	Improvement in cognitive verbal/preverbal	Severity symptoms (ADOS Communication and ADOS Social) Age at independent walking
[94] Zitter et al., 2021	Older age at intake	Learning response rate	Severity of autism symptoms Stereotyped and repetitive behaviors Adaptive behaviors MSEL DQ

^1^ Speech-related sounds/non-speech sounds. DQ: Developmental Quotient; IQ: Intellectual Quotient; MSEL: Mullen Scales of Early Learning.

**Table 7 brainsci-12-01499-t007:** Predictors of better response to EIBI and to ESDM, categorized based on number of published articles, percentage of positive studies, and combined *p*-value obtained using the Fisher’s method [55].

	EIBI			ESDM		
	Variable	N. (%) of Positive Studies	Fisher’s Statistics *	Variable	N. (%) of Positive Studies	Fisher’s Statistics *
**First-line predictors**	Higher IQ/DQ at intake	7/11 (63.6%)	χ^2^ = 83.968 (df = 20); *p* = 8.24 × 10^−10^	Verbal and non-verbal intention to communicate, attention to faces	5/6 (83.3%)	χ^2^ = 77.733 (df = 12); *p =* 1.12 × 10^−11^
**Second-line predictors**	Better receptive language abilities	2/4 (50%)	χ^2^ = 38.399 (df = 8); *p =* 6.35 × 10^−10^	Higher IQ or DQ at intake, action with objects	5/7 (71.4%)	χ^2^ = 61.444 (df = 14); *p =* 6.54 × 10^−8^
	Greater social skills	3/3 (100%)	χ^2^ = 23.799 (df = 6); *p* = 5.69 × 10^−4^	Younger age at intake	3/5 (60%)	χ^2^ = 25.633 (df = 8); *p =* 0.0012
	Communication skills	2/3 (66.6%)	χ^2^ = 17.710 (df = 6); *p* = 0.007	Less stereotyped and repetitive behaviors	2/3 (66.7%)	χ^2^ = 14.854 (df = 6); *p* = 0.021
**Weak or non-predictors**	Adaptive behaviors	2/4 (50%)	χ^2^ = 18.757 (df = 6); *p* = 0.0046	Milder severity of autistic symptoms	1/6 (16%)	χ^2^ = 22.565 (df = 12); *p* = 0.032
	Younger age at intake	1/4 (25%)	χ^2^ = 24.048 (df = 4); *p* = 7.81 × 10^−5^			
	Milder severity of autistic symptoms	1/3 (33.3%)	χ^2^ = 20.802 (df = 6); *p* = 0.002			
**Insufficient evidence**	Imitation	1/1	---	Adaptive behaviors	1/2	---
	Joint Attention	0/1	---	Imitation	0/1	---
				Joint attention	0/1	---
				Play skills	0/1	---

* d.f. = 2N; if d.f < 2N, no statistics could be retrieved from one or more original articles.

## Data Availability

Not applicable.

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
