# Peer review of "Differential Predictors of Response to Early Start Denver Model vs. Early Intensive Behavioral Intervention in Young Children with Autism Spectrum Disorder: A Systematic Review and Meta-Analysis"

_brainsci, 2022, doi:10.3390/brainsci12111499_

Round 1
Reviewer 1 Report
The present study compares response to early start Denver Model with Early Intensive behavioral intervention in young children suffering from austim spectrum disorder. The paper is well-written and of interest for the journal. However, several minor changes should be made before considering it for publication.
Abstract.
1- It is not relevant (in the abstract) that the age at first diagnosis in autism spectrum disorders is decreasing in recent years. I would focus this introduction on the response to treatment.
2- In the abstract section, the authors should describe, briefly, the methods used to carry out the systematic review.
Introduction
1- In the introduction section, second paragraph, the authors are introducing the Applied Behavioral Analysis and behavioral interventions. I recommend to start describing the overall management of these patients. What about the community mental health programs?
2- A brief introduction of the genetic background in autism spectrum disorders would be helpful.
Material and Methods
1- The methods section should be divided into several subsection. Design, data sources, study selection, assessment and measures, inclusion and exclusion criteria. Quantitative analyses, etc.
Results
1- The results section should be divided into the two groups that are to be compared. The study design of the included studies should not be described separately. I recommend to include it in the description.
2- Why are exclusion criteria reported in the results section? Are these not a priori criteria to include studies? If not, it should be discussed in the discussion section.
I recommend to add a limitations and strenghts section at the end of the discussion section. There, I recommend to include the discussion about the inclusion or exclusion criteria of the included studies.
Author Response
We thank both Reviewers for their helpful comments, which have each been addressed as specified below:
Reviewer 1
The present study compares response to early start Denver Model with Early Intensive behavioral intervention in young children suffering from autism spectrum disorder. The paper is well-written and of interest for the journal.
We thank Reviewer 1 for expressing appreciation for our work.
However, several minor changes should be made before considering it for publication.
Abstract.
1- It is not relevant (in the abstract) that the age at first diagnosis in autism spectrum disorders is decreasing in recent years. I would focus this introduction on the response to treatment.
The sentence on age at first diagnosis has been removed from the abstract.
2- In the abstract section, the authors should describe, briefly, the methods used to carry out the systematic review.
A brief description of the search strategy and inclusion criteria for eligible studies has been included in the Abstract, as follows: “Four databases (EmBase, PubMed, Scopus and WebOfScience) were systematically screened, and an additional search in the reference lists of relevant articles was conducted. Studies were select-ed if participants were children with ASD aged 12-48 months at intake, receiving either EIBI or ESDM treatment”.
Introduction
1- In the introduction section, second paragraph, the authors are introducing the Applied Behavioral Analysis and behavioral interventions. I recommend to start describing the overall management of these patients. What about the community mental health programs?
In the Introduction (page 2, par. 2), we now provide a brief description of current practices in autism treatment and management, including mental health community programs: “To date, there is no medical or biological treatment for core ASD symptoms, and interventions for ASD mainly fall within the psychoeducational, psychosocial, or behavioral frameworks [9,10]. Furthermore, there is no standard treatment for ASD. Community mental health programs, based on local guidelines, typically include a combination of interventions based on various approaches, such as speech therapy, sensory integration therapy, occupational therapy, neuropsychomotor treatment, and so on. However, these approaches are of limited efficacy and not always evidence-based, so that more structured and comprehensive interventions, like those derived from Applied Behavioral Analysis (ABA), should be preferred [9]”.
2- A brief introduction of the genetic background in autism spectrum disorders would be helpful.
A brief summary of ASD genetics has now been included in par. 1 of the Introduction, as follows: “Genetics strongly contribute to ASD, as supported by concordance in monozygotic twins being consistently higher than that observed in dizygotic twins [2]. A specific genetic aetiology is identifiable in up to 40% of individuals, although the majority of cas-es display complex gene x gene interactions involving multiple common and rare variants [3–5]. For many patients, also gene-environment interactions involving a genetic predisposition and prenatal-early postnatal environmental influences are plausible [6]. In addition to diagnosis, genetic variants can also contribute to explain interindividual variability in clinical phenotype, developmental trajectories, and responsiveness to behavioural or pharmacological treatment [7,8]. Hence, heterogeneity at the pathogenetic level translates into great clinical and treatment-related interindividual differences”.
Material and Methods
1- The methods section should be divided into several subsection. Design, data sources, study selection, assessment and measures, inclusion and exclusion criteria. Quantitative analyses, etc.
The Methods section has been divided into subsections, as requested.
Results
1- The results section should be divided into the two groups that are to be compared. The study design of the included studies should not be described separately. I recommend to include it in the description.
The Results section has been divided into par. 3.1 EIBI and par. 3.2 ESDM, as requested. For each treatment, the Study Design paragraph has been removed and its text has been moved into the general description of the selected studies.
2- Why are exclusion criteria reported in the results section? Are these not a priori criteria to include studies? If not, it should be discussed in the discussion section.
The a priori study selection criteria are indeed reported in the Methods section. In par. 3.1.1 and 3.2.1, we instead summarize the “patient selection criteria” applied in the set of studies we have reviewed. To avoid confusion, the subtitles of these sections and several sentences have been rephrased.
- I recommend to add a limitations and strenghts section at the end of the discussion section. There, I recommend to include the discussion about the inclusion or exclusion criteria of the included studies.
We generated a novel Sect. 5 “Limitations and Strengths” at the end of the Discussion, where we present the strengths and limitations intrinsic to our study inclusion and exclusion criteria.
Reviewer 2 Report
The article is presented as a systematic review of the literature, however, it contains all sections of the original study: introduction, materials and methods, results, discussion and conclusions.
This article provides recommendations for using the EIBI and ESDM models at an early age to identify factors associated with a positive response to early intervention and that may contribute to maximizing the clinical effectiveness of intervention strategies.
The Results section contains a lot of descriptive data that needs to be edited and replaced with more evidence-based, imperatively formulated formulations.
I believe that in this article it is necessary to conduct a quantitative analysis that will help to systematize the literature data.
The authors present data in tables, but these tables need to be more fully analyzed and not limited to descriptive characteristics.
The Discussion section is written constructively, although it needs editing.
In general, the work deserves a good assessment and can be useful in clinical practice, although the weak point of the work is the lack of adequate quantitative analysis.
Author Response
We thank both Reviewers for their helpful comments, which have each been addressed as specified below:
Reviewer 2
1) The article is presented as a systematic review of the literature, however, it contains all sections of the original study: introduction, materials and methods, results, discussion and conclusions.
We have structured our manuscript according to the Instructions for Authors of Brain Sciences, which recommend to follow the PRISMA guidelines for systematic reviews. According to these guidelines, a systematic review should include the following sections: introduction, methods, results, and discussion. Should the Editorial Office request formal changes in the structure of our manuscript, we shall be happy to abide by their indications.
2) This article provides recommendations for using the EIBI and ESDM models at an early age to identify factors associated with a positive response to early intervention and that may contribute to maximizing the clinical effectiveness of intervention strategies. The Results section contains a lot of descriptive data that needs to be edited and replaced with more evidence-based, imperatively formulated formulations. I believe that in this article it is necessary to conduct a quantitative analysis that will help to systematize the literature data. The authors present data in tables, but these tables need to be more fully analyzed and not limited to descriptive characteristics.
We thank Reviewer 2 for addressing this very important point. His/her comment has spurred us to perform a metanalytic combination of P-values using the classical Fisher’s method, in order to quantify the evidence presented in Table 7 for each putative predictor. Therefore in this revised version of our manuscript, table 7 now includes not only “number of studies with a significant association/total number of studies performed”, but also a cumulative P-value for each predictor. The two parameters jointly provide a much more complete picture of the extent and convergence of available evidence for each predictor. Hence this additional analysis has, in our opinion, significantly strengthened our conclusions. Our metanalytic procedure is now explained in par. 2.5 of the Methods. The title of our manuscript and one sentence of the Abstract have been modified accordingly.
3) The Discussion section is written constructively, although it needs editing.
The Discussion has been thoroughly reviewed and edited.
4) In general, the work deserves a good assessment and can be useful in clinical practice, although the weak point of the work is the lack of adequate quantitative analysis.
We believe that combined P-values, in addition to the relative number of positive studies, now provide converging evidence demonstrating different levels of strength for each predictor.
Round 2
Reviewer 2 Report
In general, the authors corrected the original version of the manuscript and conducted a statistical analysis, the results of which they presented as values ​​in Table 7. However, in order to see what changes the authors made to the text, it was necessary to provide a version of the article that shows the corrections made to text. In the revised version of the manuscript, such changes are not visible and it is the intermediate version of the article that should be requested, which shows specific changes, incl. stylistic and editorial corrections that the authors made to the original manuscript of the text.